# Medication multiple experiences of elderly patient with multiple chronic condition: A qualitative meta-synthesis

Guo Xin[1]☯, Wang Yuanxing[2]☯, Wu Jiaming[1], Hu Xue ORCID [3]*

**1** School of Nursing, Yunnan University of Chinese Medicine, Kunming City, Yunnan Province, China,
**2** Faculty of Sociology, Lomonosov Moscow State University, Moscow city, Russia, **3** Neonatology, Yan'an Hospital Affiliated to Kunming Medical University, Kunming City, Yunnan Province, China

☯ These authors contributed equally to this work.
\* huxue123451@qq.com

## Abstract

### Purpose

To determine the experience of medication multiple in elderly patients with multiple chronic condition by systematically reviewing, retrieving, and synthesizing data from qualitative studies.

### Methods

Nine databases were systematically searched for relevant contributions from the time of construction until October 30, 2024. All qualitative studies in English and Chinese exploring the real-life experiences, feelings, etc, of medication multiple in elderly patients with multiple chronic condition were included. The Enhancing Transparency in Reporting the Synthesis of Qualitative Research (ENTREQ).Two independent reviewers selected the studies and assessed the quality of each study, and the included studies were inductively analysed and interpreted then reported as a meta-synthesis.

### Results

A total of 10 studies revealed 13 sub-themes and 4 descriptive themes:Difficulties and challenges in Polypharmacy, Multi-party support facilitates Polypharmacy in elderly patients with multiple chronic condition,Positive attitude towards taking medication.

### Conclusion

The health outcomes of elderly patients with coexisting chronic diseases can be significantly impacted by polypharmacy difficulties and challenges as well as lower

**Data availability statement:** All relevant data are within the paper and its Supporting information files.

**Funding:** The author(s) received no specific funding for this work.

**Competing interests:** The authors have declared that no competing interests exist.

medication adherence.Increasing medicine knowledge among older patients with concomitant chronic diseases, bolstering medication-taking beliefs, and motivating family members to support the patient are all ways to improve medication adherence. [REGISTRATION: PROSPERO ID: CRD42024628547]

---

## 1 Introduction

As the global disease spectrum changes and the aging process accelerates, chronic diseases have replaced infectious diseases as the major threat to human health and quality of life. In 2008, the World Health Organization (WHO) defined the combination of two or more chronic diseases as "Multimorbidity," also known as chronic disease co-morbidity and multiple chronic diseases [1]. In 2016, the WHO added a definition of multimorbidity as two or more chronic diseases, long-term mental disorders, or long-term infectious diseases in the same patient [2].Many studies in recent years have shown that multimorbidity have become a major threat to health worldwide [3,4]. With the accelerating process of global population aging, the co-morbidity of multiple chronic diseases in the elderly population is increasing and is gradually becoming the norm in the health status of elderly patients worldwide [5].It has been shown [6] that elderly patients with multiple chronic diseases report severe limitations in physical functioning, loss of voluntary movement, and reduced quality of life, and have a poorer prognosis than elderly patients with a single disease.

Medication is one of the most important ways to control the condition of elderly patients with multiple chronic condition, and whether patients can take medication strictly according to the doctor's instructions greatly affects the effectiveness of chronic co-morbidities management. Poor Polypharmacy behavior can hinder the treatment and rehabilitation process, increase the readmission rate and medical costs, and even lead to deterioration of the condition and the occurrence of death. Elderly patients with multiple chronic condition often require multi-drug combination therapy, and multi-drug administration requires relatively high drug self-management behaviors in elderly patients [7].Compared to a single disease, older patients with multiple chronic condition have a greater number of prescribed medications and more complex treatment regimens, as well as greater pressure to manage their medications [8]. In addition, as older people's bodies progressively age, their cognitive and physiological functioning may decline, which makes their self-management abilities relatively weak [9].A study [10] investigated that the incidence of medication non-adherence behavior in elderly patients with multiple chronic condition ranged from 20% to 80%. Therefore, when elderly patients with multiple chronic condition use multiple medications, there are more serious problems of drug mismanagement, poor adherence, and low safety, which seriously increases the risk of death, prolongs hospitalization, and reduces the quality of life, and it is necessary to focus on the problem of their multiple medication use.Currently, there has been a gradual increase in qualitative studies on the experience of Polypharmacy in elderly co-morbid patients worldwide, but the results of a single qualitative study do not comprehensively

and accurately reflect the experience of Polypharmacy in elderly patients with multiple chronic conditon, and it is difficult to guide clinical practice. And the understanding, expression, and psychological experience of polypharmacy self-management vary among co-morbid patients from different countries, races, and cultural backgrounds. Therefore, this study aims to comprehensively collect qualitative studies related to the experience of polypharmacy in elderly patients with multiple chronic diseases, further analyze and summarize the results of the qualitative studies, and deeply analyze the experience of Polypharmacy in elderly patients with multiple chronic diseases, so as to provide a reference basis for the development of relevant nursing strategies by clinical workers.

## 2 Methods

### 2.1 Design

The systematic literature search and quality appraisal drew from the Joanna Briggs Institute (JBI) qualitative systematic review methodology [11] and qualitative meta-ethnography [12,13], Researchers adopted Noblit and Hare's 7-step approach [13] from search through to expression of the synthesis findings. and were reported according to the framework of enhancing transparency in reporting the synthesis of qualitative research (ENTREQ) [14].

### 2.2 Literature search strategies

We searched PubMed, Web of Science, Embase, Cochrane Library, CINAHL, Ebase, CNKI, VIP, CBM, and WANFANG for qualitative studies on the experience of Polypharmacy in elderly patients with multiple chronic diseases, using a combination of subject terms and free words in october 30, 2024. The search was conducted using a combination of subject terms and free words (step 1). Included:Aged, Elderly, senior, old adults; Multiple Chronic Conditions, Multiple Chronic Health Conditions, Multiple Chronic Medical Conditions, Multiple Chronic Illnesses, comorbidity; polypharmacy, polymedication, multiple medication, multiple drug, Medicine drug; medication therapy management, drug therapy management, medicines management, medication management, medicines management;qualitative research*, rooted theory, phenomenology, descriptive research, interview, experience, opinion. To ensure a comprehensive literature search, Boolean operators were used to combine search terms and manual searches and the lists of references included in the studies were screened.Take PubMed as an example of a search formula in S1 File.

### 2.3 Literature inclusion and exclusion criteria

#### 2.3.1 Inclusion criteria.

(1) **Participant (P)**

Elderly patients with multiple coexisting chronic diseases

(2) **Interest of phenomena (I)**

Real-life perceptions of the Polypharmacy process in elderly patients with multiple co-morbid chronic conditions Feelings

(3) **Context (Co)**

Hospitals, community health centers and homes of elderly patients

(4) **Study design (S)**

Different types of qualitative research, including rooted theory, phenomenology, descriptive research, etc.

#### 2.3.2 Exclusion criteria.
Duplicate publications; Incomplete information or inaccessible full-text literature; Literature not in English or Chinese; Reviews and academic papers;Literature with a C-level quality evaluation.

## 2.4 Literature screening

Two researchers with evidence-based training screened the literature and extracted information according to the inclusion and exclusion criteria. The literature was initially screened by reading the title and abstract and then re-screened by reading the full text. In case of disagreement, the decision was discussed with the third researcher. Data extraction included authors, regions, research methods, subjects, phenomena of interest, and main findings.All related articles are managed using EndNote X9 Reference Manager, which also helps to identify duplicate entries.

## 2.5 Criteria for evaluating the methodological quality of literature

Included literature was independently evaluated by two researchers using the JBI Center for Evidence-Based Health Care Quality Evaluation Criteria for Qualitative Research [15](step 2). In case of disagreement, the decision was discussed with a third researcher. A-level is to fully meet the quality standards and minimize the possibility of bias, while B-level partially meets the quality standards with a moderate likelihood of bias. C-level does not meet the quality standards at all, and there is a high possibility of bias occurring.This study only included literature with a quality level of A or B.

## 2.6 Data abstraction

The authors extracted the data, Data extraction includes the author of the literature, publication time, country, research subjects, research methods, phenomena of interest and main results, etc.and recorded them on a standardized data extraction form.Studies with no clear findings according to the extracted data were summarized according to the synthesized topic and discussion(step 3).

## 2.7 Data synthesis

This study adopted the stepwise approach proposed by Malterud [12], Noblit and Hare [13], and Sattar [16] et al, to extract qualitative research data into Microsoft Excel™ for organization and synthesis to determine the relationships among all studies (Step 4).To systematically examine and organize the data, translate studies (step 5), and synthesize translations (step 6), the experience of medication multiple in elderly patients with multiple chronic condition were analysed independently by the two reviewers. New themes and subthemes were constructed and, to express the synthesis (step 7), findings of steps 1–6 were elaborated on in this new translation and explored in relation to the wider research. Data from all 10 studies progressed to meta-synthesis. All extracted quotes were independently reviewed by two researchers, both with training and experience in qualitative research and analysis. All final themes and subthemes were determined through discussion and consensus.When two researchers disagreed, the decision was discussed with the third researcher.

## 3 Results

### 3.1 Procedure for extraction

The preliminary search yielded 1315 documents. After removing duplicates, 976 documents were obtained. Further reading of the titles, abstracts, and full texts of the documents resulted in obtaining 208 documents. These documents were selected by excluding those that did not conform to the content of the study and the subject matter, as well as those that could not be accessed in full text. Finally, after quality evaluation, 10 documents were included,The screening process is depicted in S1 Fig.

### 3.2 Basic characteristics of the included literature and results of quality assessment

These 10 studies [17–26] included a total of 204 elderly patients with multiple chronic conditions. A variety of research methods were involved, including 6 phenomenological studies, 1 descriptive qualitative study, 1 grounded theory study, and 1 mixed-methods study. The extraction results and the results of literature quality evaluation are shown in Table 1.

**Table 1. Key characteristics and quality appraisal.**

| References, year | Country | Research method | Sample size | Interest of phenomena | Results | Quality category |
|---|---|---|---|---|---|---|
| Eliott et al [17], 2007 | USA | rooted theory | 20 | How older adults with multiple illnesses make choices about medicines. | Five themes:Overview of Medicine-taking Behavior; Do People Make Choices Between Medicines; What Influences People's Choices; How Important are ComplexityandCost of Regimens; Do People Use More Than 1 Factor When Making Choices; | B |
| Haverhals et al [18], 2011 | USA | Phenomenological approach | 34 | the latent concerns and challenges faced by older adults and caregivers in managing medications. | Five themes:seeking reliable medication information;-maintaining autonomy in medication treatment decisions; worrying about taking too many medications; reconciling information discrepancies between allopathic and alternative medical therapies; and tracking and coordinating health information between multiple providers; | B |
| Meranius et al [19], 2015 | Sweden | Phenomenological approach | 28 | To explore the experience of self-managingmedication among older people with multimorbidity. | Four themes:Adapting to a new lifestyle; Ambivalence towards medicine; Experience of side effects and concerns about medical errors; Relationships as sources of feeling secure | B |
| Vander-mause et al [20], 2016 | USA | Mixed study | 30 | the experiences of older adults with multiple chronic medical conditions when a new medication was added to their existing multiple medication regimen. | One themes: preserving self | B |
| Meranius et al [21], 2016 | Sweden | Phenomenological approach | 20 | how older adults with multimorbidity experience medication self management and how this is affected by the healthcare system. | Three themes:Lack of participation in healthcare communication hinders adherence and safety; Feeling abandoned to self-care leads to health risk-taking; The healthcare organisation as an obstacle to medication self management; | B |
| Roux et al [22], 2019 | Sweden | Descriptive qualitative analyses | 19 | The medication practices and spatial dimensions of medication management for home-dwelling older adults after hospital discharge | Two themes:: the spatial dimensions of medication management in this specific context; the process of integrating medication changes into routines and familiar spaces, and the individual and collective management of medication changes linked to a renegotiation of the boundaries between public and private spaces. | B |
| Wang et al [23], 2020 | China | Phenomenological approach | 14 | Multiple medication experiences and factors influencing multiple medication use in homebound elderly patients with multiple co-morbidities | Five themes:: Higher need for medication-related knowledge; high financial stress; poor medication management practices; altered emotional perceptions; barriers to daily living; inadequate family support; | B |
| Foley et al [24], 2022 | Ireland | Phenomenological approach | 15 | the experience of self-managing multimorbidity among older adults, with a focus on medication adherence. | Six themes:seeking reliable medication information; maintaining autonomy in medication treatment decisions; worrying about taking too many medications; reconciling information discrepancies between allopathic and alternative medical therapies; and tracking and coordinating health information between multiple providers; | B |
| Yang et al [25], 2022 | China | Phenomenological approach | 21 | In-depth understanding of the real experience of the treatment burden of elderly patients with chronic diseases | Two themes:Burden of drug administration; burden of adverse drug reactions | |
| Guo et al [26], 2023 | China | Phenomenological approach | 17 | Participation in Medication Management Experiences for Older Adults with Chronic Co-morbidities | Four themes:Positive attitudes towards participation, lack of capacity for participation, lack of adaptation to the transition period, and inadequate communication of information. | B |

**Table 2. Descriptive themes, sub-themes, and frequencies.**

| descriptive themes | frequencies | sub-themes | frequencies |
|---|---|---|---|
| Difficulties and challenges in Polypharmacy [17,20–26] | 8 | Poor Polypharmacy practices [17,23–26]<br>Higher economic burden of polypharmacy [17,23]<br>Cognitive decline in the aging body [21,22,24,26]<br>Disorders of daily living due to polypharmacy [17,20,23–25].<br>Negative emotions resist medication [17,20,23,26] | 5<br>2<br>4<br>5<br>4 |
| Low level of drug literacy [17–19,21,23,26] | 6 | Medication behavioral bias [17,23,26]<br>Limited absorption of drug information [23,26]<br>Complexity of multi-physician management [18,19,21] | 3<br>2<br>3 |
| Multi-party support facilitates Polypharmacy in older patients with coexisting chronic conditions [17,19,20,22–24,26] | 7 | Self-support [17,19,20,22,26]<br>Family support [22–24]<br>Social support [19,20,23,26] | 5<br>3<br>4 |
| Positive attitude towards taking medication [20,22,23,26] | 4 | Proactive access to medication information [20,23,26]<br>Proactive response measures [20,22] | 3<br>2 |

## 3.3 Thematic synthesis

A total of 4 descriptive themes were synthesized in this study, and these themes further included 13 sub-themes, as shown in Table 2.

## 3.3 Themes

### 3.3.1 Meta-Theme 1: Difficulties and challenges in Polypharmacy.

**Sub-theme 1: poor polypharmacy practices:** Elderly patients with multiple chronic condition are prone to a variety of problems such as low adherence and poor medication management because of the coexistence of multiple diseases and the need for long-term treatment with multiple medications [17,23–26]. The complexity of medication types, dosing times, and frequency can affect patients' medication management, and the more complex the patient's medication regimen, the heavier the medication management burden. The dosage form, size, and flavor of medications are also likely to negatively impact patient medication practices. Some special medications bring a greater management burden to elderly patients in terms of preservation, for example, insulin needs to be preserved at low temperatures, which is not easy for patients to carry outside the home and can easily affect their quality of life. Some elderly patients buy large quantities of medications in one visit due to the long distance to the healthcare facility, which makes them susceptible to taking expired medications. In addition, the complexity of medication regimens to mitigate adverse drug reactions may further increases the burden of medication management experienced by elderly patients with multiple chronic conditon.

*"My biggest frustration with the doctors is overdosing...Here a couple years ago I ended up in the hospital because I was getting bad dizzy spells and I couldn't breathe and after they checked me out they found out they had overdosed me with medications—the doctor had. So they had to readjust my blood work and medications all over again." [17]*

*"Every time I drink that pill, especially with a bad smell, a burp of that flavor can make me want to vomit; therefore, every time I take medicine I am afraid." [23]*

*"Now I've got rheumatoid arthritis in the wrists and I'm on methotrexate once weekly where I take eight of them every Wednesday and to counteract those I've a folic acid to take once a week, every Thursday, the day after I take my methotrexate." [24]*

*"Taking insulin is inconvenient compared to normal. Everywhere you go, you have to pick it up, especially in the summer it's annoying. When the temperature is high, you have to have an ice pack, or it only stays that long. It's a pain in the ass."* [25]

*"The types of medication used are many and complex. Every day I have to deal with a bunch of medications, and I've made several mistakes, so I'll have to organize and classify them when I have time."* [26]

**Sub-theme 2: higher economic burden of polypharmacy:** Due to the different types of diseases among elderly patients with multiple chronic disease, medication treatment programs are more complex than those for patients with a single disease, and medication costs are relatively high [17,23]. At the same time, there are differences in reimbursement rates for hospitalization for patients with different types of health insurance. For those patients who do not have health insurance, do not have medication chronic disease benefits, or do not have insurance, financial hardship is an important reason why patients have a poorer experience with multiple medications. In addition, the increased cost of medications predisposed respondents to change their treatment regimen or refuse medications altogether.

*"I said my God, what do they think I'm a bank here or what! I have a lot of pills. But if they [prices] jump like that I'll have to start cutting them in half or stop taking them. So a few of them that I haven't been taking I don't take them anymore. Just can't afford it."* [17]

*"Every time you take a big dose of medicine, it's not just medicine, it's all money, it's all the kids' hard-earned money (helpless, frustrated expression)."* [23]

**Sub-theme 3: cognitive decline in the aging body:** With the increase of age, all the functions of the elderly patients are declining, and at the same time, they are troubled by many kinds of diseases. Most of the elderly patients' physical and mental state will change [21,22,24,26]. They are not easy to accept new things, and their comprehension, attention, and memory will decline to varying degrees. Lack of self-management of drug use, patients are prone to forget to take medication, omit to take medication, miss the best time to take medication, and so on. There is often improper use of drugs, poor compliance, and other phenomena, reducing the effectiveness of drugs and increasing the incidence of adverse drug reactions.

*"I was really on top of my medication and I was able to say I have this pill for this and that pill for that. But I can't anymore."* [21]

*"Because now, I've got all my medication out in one box, all the medication that I take. That way, there is no mixing them up with the others."* [22]

*"He said to me not to take the other tablets with it at the time being. That they wouldn't be – it wouldn't be as effective. So I do leave them until – I take them with my lunch, you know? And when I do that I forget to take them entirely if I don't leave them on the table in the morning. "* [24]

*"Hearing is failing and reactions are slow, speak louder and more slowly a few times."* [26]

**Sub-theme 4: disorders of daily living due to polypharmacy:** Adverse drug reactions are prone to disrupt the daily rhythms of elderly patients with multiple chronic disease, and the disruption of these rhythms will adversely affect the physical condition of elderly patients and even aggravate the adverse reactions of their conditions [17,20,23–25]. At the same time, elderly patients with multiple medications need to carry their medications with them when they go out and take their medications at the specified time points. This not only affects the convenience of going out, but also increases the medication shame of elderly patients and reduces the quality of social activities.

*"The reason I don't like to take it [diuretic], because I have to go and urinate so many times a day, it gets so discouraging, and it gets so strong, the feeling that if I ever left the house I'd be in trouble."* [17]

*"My A1C was in the normal range! So I celebrated and made a juicy cake!But I beat myself up because I went against my diet plan and it was food I couldn't eat。"* [20]

*"The last time the doctor gave me that medicine prescription to take before bed, I ended up spending the night in the bathroom. Originally, I had a bit of neurasthenia, which caused me to have a sleepless night. The next day, my blood pressure was high - ugh."* [23]

*"Once the old man went out to eat with me, I have diabetes and have to take insulin. Before the meal, I must take a shot. On many occasions, I feel strange and embarrassed. The old man says nothing, but I rarely go out to eat with them. That kind of feeling can be bad."* [23]

*"Again, the inhalers would be the top priority on my list. If by any chance the inhaler - I usually keep it in the bedroom, both of them in my bedroom - and if I go downstairs in the morning and I don't use the brown one, I will actually go upstairs and take it because I live in fear of running into COPD."* [24]

*"Metformin is very damaging to the stomach, and when I take this medication, my stomach gets upset, rises, and I can't eat."* [25]

**Sub-theme 5: negative emotions resist medication:** Due to the length of their illnesses, the strain of taking numerous drugs over time, and the significant financial load, elderly individuals with multiple chronic disease experience unpleasant feelings like anxiety, pain, fear, shame, and even social stigma [17,20,23,26].

*"I've tried to stop taking those tablets [carbamazepine] for a while and I got a really bad pain in my face and it made me cry and I went back to my doctor and he said you're stupid. Don't stop taking those medicines."* [17]

*"You look at all the bottles up there and you justshakeyourhead."* [20]

*"In a rural family, my partner has left and my children are running a small business here. They don't make much money, so in addition to feeding and clothing me, they have to spend money on my medication, so I feel quite sorry for them."* [23]

*"When all you do is take pills all day long, aside from eating, what's the point of living?"* [26]

**3.3.2 Meta-theme 2: low level of drug literacy. Sub-theme 1:medication behavioral bias:** Due to a variety of factors such as side effects of medications, cost, comfort, and patients' subjective concept of medication, some elderly patients with multiple chronic disease may have deviations in medication behavior, such as stopping medication without authorization and arbitrarily adjusting the dosage and frequency of medication use [17,23,26].

*"The reason I don't like to take it [diuretic], because I have to go and urinate so many times a day, it gets so discouraging, and it gets so strong, the feeling that if I ever left the house I'd be in trouble."* [17]

*"I often forget to take the medicine, sometimes feel no discomfort, so I do not eat. When it is difficult, I then take it quickly. After all, the medicine is three times poison."* [23]

*"Take an extra tablet when you feel serious, skip it occasionally for a few days when you're better, and take it again when you're serious."* [26]

**Sub-theme 2:limited absorption of drug information:** Cognitive decline with age affects the acquisition, understanding, absorption, and application of drug information in elderly patients. Moreover, elderly patients with multiple chronic disease

often need long-term treatment with multiple medications and have complex medication regimens [23,26]. Complicated medication instructions and different dosing schedules make elderly patients with multiple chronic disease unable to absorb and internalize medication information, which leads to a decrease in medication adherence.

*"I'm getting old and my memory is poor. I often forget to take my medication and have to be reminded by my children before I remember."* [23]

*"Remember it now and forget it in a few days."* [26]

**Sub-theme 3: complexity of multi-physician management:** When multiple physicians are involved in medication management, elderly patients with multiple chronic disease find it difficult to keep track of the medications they take and what they are used for, which makes them feel powerless and confused, increases their fear of medication, and even leads to a reluctance to take their medication on time or adjust the dosage on their own [18,19,21]. Particularly if the doctor fails to clearly explain the cause of the illness and the effects of the treatment, the patient may stop taking the medication because he or she believes that the medication is not important.

*"I see two different doctors on a regular schedule and the only thing I really worry about is if one of them changes my medicine I want the other one to know about it because I get medicine from each of them and I don't want to add something or take away something that is going to cause problems."* [18]

*"I received a letter from my GP saying she has been in contact with the rheumatologist and cardiologist, and she cannot understand why they have taken me off Indomee."* [19]

*"I think it's a matter of prestige between the doctors. They don't like to consult each other. The family doctor is not willing to make contact with the kid ney specialist, nor is the cardiologist, who prescribed inappropriate tablets even though I asked him to consult the kidney specialist."* [21]

**3.3.3 Meta-theme 3: multi-party support facilitates polypharmacy in older patients with coexisting chronic conditions. Sub-theme 1:self-support:** Some elderly patients with multiple chronic disease demonstrate positive coping attitudes by adjusting daily routines to accommodate medication-taking schedules, or by using medication boxes to manage medications, or by shifting attention from medication-taking behaviors to more meaningful activities [17,19,20,22,26]. They look for personal success in the medication-taking process and strive to keep polypharmacy positive and active. In addition, elderly patients with multiple chronic disease try to obtain information about medication management through a variety of channels, such as medication instruction manuals, telephone counseling, or Internet searches.

*"She gave me some stretching exercises that seemed to help an awful lot. I was able, finally, not to take the [celecoxib] and still get along pretty well."* [17]

*"The only thing that bothers me is my eyes because I love reading. But I don't care about anything else as long I have my books."* [19]

*"have a pill box for morning, noon and evening. Every Sunday morning, I organize everything so that it is much easier to take my daily medication."* [20]

*"I take my medication in the kitchen in the morning, before breakfast."* [22]

*"Don't look at our age, we also use smartphones well, small programs, and the Internet."* [26]

**Sub-theme 2:family support:** At present, the number of elderly patients with multiple chronic disease, a special group of patients who often face mental and psychological distress during the prolonged use of multiple medications, is increasing

[22–24]. They desperately want the care, help, and companionship of their spouses and children, as family members offer significant benefits in helping patients with self-care and proper medication use. In addition, the concurrent medication-taking behavior of family members can serve as a reminder and support for each other.

*"But when she takes hers, I see it, and I say "Hey! I'd better take mine." If I take mine, she says "Oh! I've got to take them too."* [22]

*"I live with my daughter, who is a nurse and reminds me to take my medication every day. I haven't been sick for so many years, thanks to my daughter."* [23]

*"I've a good wife, fair play to her. She puts the drops in me [sic] eyes and apart from that 'tis only a matter of taking the tablets. She'll look after me."* [24]

**Sub-theme 3:medical support:** Elderly patients with multiple chronic disease receive correspondingly less medical support after discharge from the hospital and move from a state of being fully cared for to a state of self-care in a short period of time; the rapid role change is prone to negatively affect their polypharmacy behaviors, and the patients expect to have channels of communication and feedback even after discharge from the hospital [19,20,23,26]. Physician assistance is closely related to the polypharmacy ability of elderly patients with multiple chronic disease. During their daily contact and communication with patients, doctors can not only provide patients with different types of health counseling, health education, and health assessment services but also teach patients disease-related knowledge and precautions for medication use, and urge them to take medication on time. Therefore, high-quality doctor-patient communication can not only promote patients' medication adherence but also improve their polypharmacy ability.

*"I've known the nurses for a long time and they always explain the medications for me. They take good care of me and I can always ask them if I have some questions or concerns about the medications I take."* [19]

*"I feel a kinship with my physician."* [20]

*"The last time I had a stomachache, my daughter gave me the Internet to find out if the side effects of these drugs, that the Internet is a mess, say what there. I do not dare to believe; I still believe that the big hospital doctor's"* [23]

*"I expect that there will still be avenues for communication and ongoing follow-up on medications after discharge."* [26]

**3.3.4  Meta-theme 4:positive attitude towards taking medication. Sub-theme 1:proactive access to medication information:** Some elderly patients with multiple chronic disease will take the initiative to have face-to-face consultations with the healthcare team when they are faced with doubts about medication use (e.g., precautions for medication use, how to recognize adverse reactions, how to cope with unexpected situations, etc.) [20,23,26]. As the elderly patient seeks information, takes the initiative to track his health status, and adjusts his medication strategy accordingly, his self-confidence gradually increases. He begins to face reality, providing comfort and encouragement to himself, as well as being filled with gratitude and always maintaining a positive attitude towards taking his medication on time.

*"These are all very conscious daily decisions that you have to make when you take multiple medicines.... If you want to look at it, it's a pain in the neck, but, you know, it's keeping me going and I feel very good because of it"* [20]

*"My son told me that he had seen a lot of people with similar illnesses to mine who were cured, so I thought, "If others can be cured, why can't I?" I have to stick to the medication, even for the sake of my child."* [23]

*"I tell my doctor about the medications I'm not comfortable with and ask for them to be adjusted to fit my personal habits so I'm better able to stick to them."* [26]

**Sub-theme 2:proactive response measures:** Some older multimorbid patients demonstrate positive coping attitudes by adjusting their daily routines to accommodate the timing of their medications or using medication cartridges to manage their medications.

*"I do the regimen the way I figure it will fit, trying to balance them on the 12-hour schedule."* [20]

*"The weekly pillbox is always there, in the same place. Ah! That stays right there, always on my table."* [22]

## 4  Discussion

### 4.1  Elderly patients with multiple chronic conditions have poor the experience of medication multiple and should focus on providing adjunctive support

With the increase in the prevalence of multiple chronic condition in the elderly, the phenomenon of elderly chronic disease coexisting patients applying multiple medications at the same time is becoming increasingly common. Patients with more types of medication have longer medication cycles, and their medication burden has also increased. The medication burden is an important factor affecting the patient's ability to self-manage their medications [27].In the early stage of polypharmacy, the development of polypharmacy ability of elderly patients is affected by a combination of insufficient information about medications, complex medication regimens, and their own medication literacy. In the middle stage of polypharmacy, elderly patients' ability to self-manage their medications is mainly affected by their economic level, adverse effects, and barriers to daily living. In the late phase of polypharmacy, elderly patients are more likely to have negative treatment outcomes due to the heavy financial burden of long-term medication use, persistent poor medication management ability, and worsening negative emotions.We should instruct elderly patients to classify and organize their medications in an orderly manner, and we also need to provide them with specialized training in medication management skills so that they can rationally arrange their daily medication schedules and further enhance theirmedication adherence. In addition, taking into account the decline in physiological functions such as cognition, memory, and action of elderly patients, we can adopt methods such as concise and easy-to-understand written teaching, rich graphic descriptions, or popular science animation to help patients better understand and absorb drug-related information. The government and society should accelerate the improvement of the medical insurance mechanism for chronic diseases in the elderly, and strengthen the relevant medical laws, regulations, and policies to ensure that elderly patients with multiple chronic condition receive appropriate support for their medical expenses, so as to reduce their financial pressure.

### 4.2  Strengthening guidance and supervision to improve medication adherence in elderly patients with multiple chronic diseases

The results of this study found that, influenced by the side effects of drugs, cost, comfort, and patients' subjective concepts of medication, when faced with complex medication regimens, elderly patients with multiple chronic condition are prone to negative and ambivalent medication moods and incorrect medication behaviors, which can easily lead to a decrease in the level of adherence to their medication regimens.In addition, when multiple physicians are involved in prescribing, older adults may feel confused about who is responsible for their medication management. This lack of continuity and clarity can increase their anxiety and affect their medication adherence. Decreased medication adherence is a serious problem, especially for elderly patients with multiple chronic condition who require prolonged or lifelong medication. Clinical workers should increase health education for elderly patients with multiple chronic condition, so that patients and their families fully understand the role of drugs, side effects, the necessity of taking medication as prescribed by the doctor, and the harm of stopping medication without authorization, to further standardize their medication behavior, and constantly encourage patients to take the correct medication behavior as prescribed by the doctor. In the health education

of patients, the cognitive acceptance ability of the elderly group should be taken into account, to ensure that the content of the education is simple and easy to understand, scientific, and effective, to build a good doctor-patient relationship.

### 4.3 Promoting multi-stakeholder support to enhance the experience of medication multiple in older patients with multiple chronic conditions

Elderly patients with multiple chronic condition have poor medication management practices and desire support and assistance from multiple sources, including spouses, children, and healthcare professionals. Through the continuous care and supervision of family members, as well as the assistance of hospital medical staff and out-of-hospital multifaceted counseling pathways, the psychological status of patients and the final outcome of medication therapy can be effectively improved. In the process of drug treatment, the patient's trust in the medical staff and whether the patient feels that the medical staff pays attention to the disease he or she is suffering from will affect the patient's ability to self-manage the drug. Therefore, it is suggested that clinical medical workers should improve their own professionalism and enhance the communication between patients and doctors so that patients can understand their own diseases and medication programs. In addition, with the increasing popularity of digital technology, hospitals and society should provide a variety of options, such as online medical platforms, to meet the communication needs of elderly patients, so that patients can communicate their health needs at home through electronic means, such as WeChat and applets, avoiding unnecessary medical treatment and saving time and travel costs.

### 4.4 Promoting positive attitudes toward medication taking and improving medication adherence in elderly patients with multimorbidities

Elderly people with chronic illnesses who can deal with the difficulties of taking several drugs with a positive outlook have better medication adherence rates and can more quickly adapt their physical and mental states to the difficulties and challenges that come with taking medication. A positive emotional state also makes it easier for patients to ask for aid and support from friends and family, make healthier lifestyle choices, and engage in health-promoting behaviors like taking their prescriptions as directed by their doctors [28]. As a result, we should help older patients with coexisting chronic illnesses feel happy, support them in pursuing healthy interests and activities, take the initiative to find out about pertinent medications, and keep them feeling upbeat and hopeful.

## 5 Limitations

This study has several limitations.First, according to the literature's inclusion criteria, only qualitative research that was published in English- or Chinese-indexed journals was chosen. Because certain gray literature was not searched, the results could be skewed. At the same time, The quality of the included literature was rated as B grade. Furthermore, due to cultural and policy variations across nations and regions, the definition and details of polypharmacy experience in elderly patients with multiple chronic conditions may change. Of the 10 papers included, 8 did not mention the influence of the researcher's own values and cultural background.The psychological requirements and experiences of older patients with coexisting chronic conditions in terms of drug management are significantly influenced by the cultures of various locations, which may result in varied outcomes. Different researchers have different understandings and interests.There is a need to include high-quality studies in the future to enrich the results of this study.

## 6 Conclusion

This study used a qualitative research systematic integration and analysis approach, which, to some extent, integrated studies from different countries and medical backgrounds, and more realistically reflected the true psychological experience of polypharmacy in elderly patients with coexisting chronic diseases.The results of this study show that elderly patients with coexisting chronic diseases have psychological experiences such as difficulties and challenges

in polypharmacy, changes in emotional cognition, lower levels of medication adherence, and the need for multi-party support. In the future, healthcare professionals can start by strengthening patients' knowledge of medications, enhancing the self-management skills of medications in elderly patients with coexisting chronic diseases, while focusing on in-depth communication and timely psychological adjustment with elderly patients, mobilizing resources from all parties to jointly promote patients' participation in the management of medications and ensuring the standardization of medication behaviors. Healthcare organizations should pay great attention to this phenomenon, improve the chronic disease health insurance system, and strengthen the role of family and social support, so as to gradually improve patients' medication experience and quality of life.

## Supporting information

**S1 File. Search strategy.**
(DOCX)

**S1 Fig. Flow chart of literature screening.**
(TIF)

## Author contributions

**Conceptualization:** Guo Xin, Wang Yuanxing.

**Data curation:** Guo Xin, Wu Jiaming.

**Formal analysis:** Guo Xin, Wang Yuanxing, Wu Jiaming, Hu Xue.

**Methodology:** Guo Xin, Wang Yuanxing.

**Project administration:** Guo Xin, Wang Yuanxing, Hu Xue.

**Supervision:** Guo Xin, Wang Yuanxing, Hu Xue.

**Writing – original draft:** Guo Xin, Wang Yuanxing, Wu Jiaming, Hu Xue.

**Writing – review & editing:** Guo Xin, Wang Yuanxing, Wu Jiaming, Hu Xue.

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
