## [Decision Letter · Decision Letter 0]

6 Aug 2025

Dear Dr. Xue, 

Thank you for submitting your manuscript to PLOS ONE. After careful consideration, we feel that it has merit but does not fully meet PLOS ONE’s publication criteria as it currently stands. Therefore, we invite you to submit a revised version of the manuscript that addresses the points raised during the review process.

We look forward to receiving your revised manuscript.

Kind regards,

Ramesh Athe, PhD

Academic Editor

PLOS ONE

Journal Requirements:

2. During your revisions, please note that a simple title correction is required: Please include the term "Scoping review" in the title. Please ensure this is updated in the manuscript file and the online submission information.

Reviewers' comments:

Reviewer's Responses to Questions

**Comments to the Author**

1. Is the manuscript technically sound, and do the data support the conclusions?

Reviewer #1: Partly

Reviewer #2: Yes

2. Has the statistical analysis been performed appropriately and rigorously?

Reviewer #1: N/A

Reviewer #2: Yes

3. Have the authors made all data underlying the findings in their manuscript fully available?

Reviewer #1: Yes

Reviewer #2: Yes

4. Is the manuscript presented in an intelligible fashion and written in standard English?

Reviewer #1: No

Reviewer #2: Yes

Reviewer #1: Thank you very much for entrusting me with reviewing the article. The subject of the study is very valuable.

The main problem with this article is the methodology used. As you know, meta-synthesis and meta-analysis are very different in nature, and the procedures and methods are different. In principle, using a registration protocol in PROSPERO is suitable for systematic reviews and meta-analyses, and is not recommended for meta-synthesis studies

Nolbit :meta ethnography

critical interpretive synthesis

framework synthesis

are the most common approaches for meta synthesis .

And every one of them has different steps .

Reviewer #2: The publication is sound and logically organized, written with standard academic English. The data analysis was clearly spelt out at the data analysis part of the paper and the findings explicitly annotated. Overall, the paper is excellent with no obvious breach of research ethic

**Do you want your identity to be public for this peer review?** For information about this choice, including consent withdrawal, please see our Privacy Policy

Reviewer #1: No

Reviewer #2: **Yes:** OBED AIDOO

---

## [Author Response · Author response to Decision Letter 1]

19 Aug 2025

Response to Reviewers

Dear Editor and Reviewers:

Greetings!

First of all, on behalf of all the authors, please allow me to extend our sincerest thanks to you and the reviewers. We would like to thank you for taking the time out of your busy schedules to review our paper in detail and provide valuable comments and suggestions. We are fully aware of the importance of these comments in improving the quality of the paper and ensuring academic rigor.

After receiving the review comments, we immediately organized our team to conduct in-depth discussion and analysis, and revised and improved the paper in strict accordance with the reviewers' suggestions. Editorial comments are listed below in red font. Reviewer comments are in blue font, such as below, where specific questions are numbered. Our responses are given in black font. The following are our responses and explanations to the reviewers' comments:

Editorial comment

1.Please ensure that your manuscript meets PLOS ONE's style requirements, including those for �le naming:

In order to comply with the strict formatting requirements of PLOS ONE, we have revised the formatting of the previous manuscript, which now conforms to PLOS ONE style, as described under "Revised Manuscript with Track Changes" and "Manuscript."

2.During your revisions, please note that a simple title correction is required: Please include the term "Scoping review" in the title. Please ensure this is updated in the manuscript file and the online submission information.

Thank you for your meticulous review of our manuscript and your valuable suggestions. However, we found that the title of the manuscript was different from that of the information submitted online. We made corrections to this, and ensure this is updated in the manuscript file and the online submission information. Regarding your suggestion to include the term "Scoping review" in the title, we have carefully considered it and held an internal team discussion. After careful evaluation, we believe that the current topic "Medication Self-Management Experiences of Elderly Patient with Multiple Chronic conditon: "A qualitative meta-synthesis" can most accurately reflect the core content and innovation points of this study for the following reasons:

Firstly, Meta-synthesis aims to integrate qualitative research data to generate new theories or in-depth explanations that explain the meaning, experience or social processes behind phenomena, and it needs to answer clear qualitative research questions. A "Scoping review" involves a standardized and systematic search and screening of literature, integrating existing knowledge on a certain topic to determine the main concepts, theories, sources, research types, and knowledge gaps of that topic. It does not require answering specific questions. However,this study aims to comprehensively collect qualitative studies related to the experience of medication self-management in elderly patients with multiple chronic diseases, further analyze and summarize the results of the qualitative studies, and deeply analyze the experience of medication self-management in elderly patients with multiple chronic diseases, so as to provide a reference basis for the development of relevant nursing strategies by clinical workers. Therefore, the phrase "A qualitative meta-synthesis" used in the title is more suitable to be the keyword used in this researce and can effectively assist readers in retrieval. Secondly, the current title is strictly consistent with the methodology section in the manuscript. We have checked the title structures of two similar studies (such as Jarden RJ, Cherry K, Sparham E, et al. Inpatients' experiences of falls: A qualitative meta-synthesis. J Adv Nurs. 2025;81(1):4-19. doi:10.1111/jan.16244; Wang Z, Shi Q, Zeng Y, Li Y. Experiences and perceptions of self-management in people with prediabetes: A qualitative meta-synthesis. J Clin Nurs. 2023;32(17-18):5886-5903. doi:10.1111/jocn.16713), and the current format conforms to the field conventions.

We fully understand your revision suggestions for enhancing the visibility of the manuscript, which has been of great help to us. We also respect your and the journal's professional decisions. If further adjustments are needed, please let us know at any time.

3.If the reviewer comments include a recommendation to cite specific previously published works, please review and evaluate these publications to determine whether they are relevant and should be cited. There is no requirement to cite these works unless the editor has indicated otherwise. 

Thank you very much for your valuable comment. The reviewers did not ask us to cite specific previously published works, so we did not quote the relevant literature of the reviewers in the manuscript.

Reviewer 1:

Regarding Reviewer 1's comments, we are very grateful to the reviewer for taking the time to read our paper. We would like to thank you for your professional review work, constructive comments, and valuable suggestions on our manuscript. Your time and efforts are greatly appreciated. We have revised the manuscript according to your comments as listed in detail below.

1. Thank you very much for entrusting me with reviewing the article. The subject of the study is very valuable. The main problemwith this article is the methodology used. As you know, meta-synthesis and meta-analysis are very different in nature, and the procedures and methods are different. In principle, using a registration protocol in PROSPERO is suitable for systematic reviews and meta-analyses, and is not recommended for meta-synthesis studies. Nolbit: meta ethnography,critical interpretive synthesis framework synthesis are the most common approaches for meta synthesis. And every one of them has different steps.

We sincerely thank the reviewers for their expertise in qualitative methods and your valuable insights into the comprehensive plan. Your comment helped us clarify the methodological nuances.We have incorporated explanations based on your recommendations to enhance clarity, and the modified content is as follows:

1�Regarding PROSPERO registration:

We acknowledge that PROSPERO is traditionally associated with systematic reviews of quantitative studies.However, after reviewing relevant literature, we found that in recent years, the official Cochrane guidelines have explicitly supported the registration of qualitative synthetic studies in PROSPERO to ensure the transparency of the methods [.Noyes J, Booth A, Cargo M, Flemming K, Garside R, Hannes K, Harden A, Harris J, Lewin S, Pantoja T, Thomas J. Cochrane Qualitative and Implementation Methods Group guidance series-paper 1: introduction. J Clin Epidemiol. 2018 May;97:35-38.doi: 10.1016/j.jclinepi.2017.09.025.], and PROSPERO now accepts registrations for qualitative evidence syntheses (QES), including meta-synthesis and meta-ethnography [PROSPERO (2023) Inclusion of qualitative evidence syntheses (QES) in PROSPERO.National Institute for Health Research (NIHR)].

In addition, our registration focus is on documenting: ① Search strategies and inclusion criteria (to avoid selection bias); ② Analytical framework (Noblit's meta-ethnography); ③ Ethical considerations of interpretive synthesis. Registration in compliance with the qualitative evidence synthesis in PROSPERO

(2)On methodology specification:

We sincerely thank the reviewers for their profound insights into our comprehensive research methods. Based on the suggestions, we conducted a rigorous examination of the methodology and found that there were indeed significant problems. Subsequently, the research team held a discussion and decided to revise the methodology section to be fully consistent with the meta-ethnographic framework of Noblit & Hare (1988)�For specific revisions, please refer to the methodology section on pages 5 to 8 of the manuscript.

We thank the reviewer for pushing us to elevate methodological rigor. These revisions will strengthen the manuscript's scholarly contribution.

Reviewer 2:

Regarding Reviewer 2's comments, we are very grateful to the reviewer for taking the time to read our paper, and it is an honor to receive Reviewer 2's recognition and praise of our paper. In addition, our careful reading of Reviewer 2's comments does not appear to suggest any changes, and no attachments were found in the emails or manuscript recordsso we have not revised Reviewer 2's comments.

---

## [Editor Report · Decision Letter 1]

21 Aug 2025

Medication Multiple Experiences of Elderly Patient with Multiple Chronic condition: A qualitative meta-synthesis

PONE-D-25-02450R1

Dear Dr. Xue,

We’re pleased to inform you that your manuscript has been judged scientifically suitable for publication and will be formally accepted for publication once it meets all outstanding technical requirements.

Kind regards,

Ramesh Athe, PhD 

Academic Editor
---

## [Editor Report · Acceptance letter]

PONE-D-25-02450R1

PLOS ONE

Dear Dr. Xue,

I'm pleased to inform you that your manuscript has been deemed suitable for publication in PLOS ONE. Congratulations! Your manuscript is now being handed over to our production team.

Kind regards,

on behalf of

Dr. Ramesh Athe

Academic Editor

PLOS ONE